# Manufacturer invasion and online sales mode strategy considering the level of service quality

**Guanxiang Zhang, Xinping Liu, Guiping Su*

Department of Electronic Business, South China University of Technology, Guangzhou, China

* ebsugpcassia@mail.scut.edu.cn

## Abstract

This study investigates the decision process of own-brand intrusion by contract manufacturers and their selection of invasion sales modes under the consideration of service quality disparities between brand manufacturers and contract manufacturers. Specifically, the study constructs a three-tier supply chain system comprising a brand manufacturer, a contract manufacturer, and an e-commerce platform. The equilibrium profits under different sales mode combinations are determined by using reverse induction methodology, and the optimal sales mode combinations are analyzed and compared. The study reveals that the decision process of contract manufacturers' own-brand invasion depends on the potential market demand. Furthermore, when brand manufacturers adopt the reselling mode, the service quality level does not affect the decision process of invasion sales modes. However, when brand manufacturers adopt the agency mode, contract manufacturers with low service quality levels are more suitable for invasion through the agency mode, whereas contract manufacturers with high service quality levels are better suited for invasion through the reselling mode. Additionally, for the equilibrium sales mode combination among members of the supply chain, it is observed that with lower commission rates, both brand manufacturers and contract manufacturers choose the agency mode, while with higher commission rates, both choose the reselling mode. When commission rates are moderate, brand manufacturers prefer the agency mode, whereas contract manufacturers prefer the reselling mode.

## 1. Introduction

With the popularity of smartphones and the impact of COVID-19, people are increasingly inclined to online consumption even in the post-COVID-19 pandemic era [1]. It's reported that the online retail sales in the United States exceeded 1 billion dollars in 2022, with a market penetration rate exceeding 20%, indicating the e-commerce market steady growth [2]. Against this background, manufacturers actively cooperate with e-commerce platforms (such as Amazon, eBay, JD.com and Tmall.com) to sell their products through online channels in order to improve their corporate profits [3]. At present, mainstream e-commerce platforms mainly provide two sales modes for sales products, namely reselling mode and agency mode. The former is mainly represented by JD's own flagship store. In the reselling mode, the goods are sold

**Funding:** This research is supported by Humanities and Social Sciences Research Project of the Ministry of Education of the People's Republic of

China [grant number 19YJA630107], Guangdong Office of Philosophy and Social Science [grant numberGD20CGL46], and 2023 Annual Project of Guangzhou Philosophy and Social Science Planning [grant number 2023GZYB20]. The funders had no role in study design, data collection and analysis, decision to publish, or preparation of the manuscript.

**Competing interests:** The authors have declared that no competing interests exist.

by the manufacturer to the e-commerce platform, which is responsible for the final sale of the goods. Ownership of the goods is transferred to the e-commerce platforms. Taobao is an e-commerce platform that focuses on agency sale. Under the agency mode, the e-commerce platform only provides a commercial platform and charges commissions which is not responsible for the sale of the goods, and the manufacturer is responsible for the entire operation and sales of the goods. At the same time, with the intensification of global market competition and the rapid development of the service economy, traditional manufacturers are experiencing a continuous compression of their profit margins. An increasing number of manufacturers are endeavoring to expand their operations into the service sector. For instance, Apple Inc. derives as much as one-fifth of its total revenue from service-related income [4]. Competition among enterprises has transitioned from single-product competition to competition between product-service supply chains. This shift is particularly evident in e-commerce platforms. When consumers make online purchases, the quality (or utility value) of the products cannot be directly perceived. Instead, it can be assessed and evaluated based on service quality. Consumers are not merely acquiring products; they are purchasing services. This type of service refers to the services that accompany the entire process of consumers purchasing a product. This type of service includes not merely post-purchase services, such as post-sales maintenance services for electronic products, which can vary in quality across different online stores. Additionally, it also comprises pre-purchase services aimed at promoting and marketing products, such as live-streamed product promotions and product usage explanations, etc. These factors underscore the increasing emphasis placed on product-service quality by businesses.

Due to increasingly specialized division of labor, many brand manufacturers have collaborated with contract manufacturers to handle the production and manufacturing processes of their branded products, allowing them to focus more on product development and upgrading brand services. The practice of outsourcing production in labor-intensive industries has become quite common. For example, Apple's manufacturing and production in China are handled by Foxconn Group [5]. In recent years, it was found that the profit margins from producing the own branded products far exceed those from pure contract manufacturing. Additionally, there is still a risk of losing business for contract manufacturers while contract manufacturing business is generally stable. Due to these reasons and more, contract manufacturers that used to solely provide manufacturing services for brand manufacturers have begun exploring the option of transformation and upgrading by creating their own independent brands. This is known as expanding their OBM (Original Branding Manufacturing) business. For example, Samsung is a significant contract manufacturer for Apple, producing OLED displays and chips for iPhone, iPad, and Mac. However, Samsung also develops its own smartphone brand, GALAXY, and introduces its products to the market [6]. A closely related example to this study is the case of the Korrun Co., Ltd, which manufactures travel luggage for Xiaomi Group as a contract manufacturer while also establishing its own brand 90 FUN and achieving tremendous success in online sales [7]. These own-branded products are sold through the "JD.com Self-Operated Flagship Store" (reselling mode) or the "Official Brand Flagship Store" (agency mode) provided by JD.com [8]. However, establishing their own brand also comes with risks. Creating a brand of their own may result in conflicts of interest with major clients, potentially leading to a decrease in contract manufacturing business and a decline in profits. Promoting, operating, marketing, and product development after establishing a brand pose significant challenges for the company that wants to build the own brand. There are numerous examples of brand establishment failures. Additionally, due to the long-standing dominance of traditional brand distributors in offline channels, most startup brands tend to choose online channels as their primary operational channels. Therefore, whether a contract manufacturer aspiring to build its own brand should establish a brand and choose the

right sales mode on e-commerce platforms becomes a crucial question, which is the focus of this research.

Under this background, based on real-world scenarios, this study establishes a three-tier supply chain game-theoretic model. Considering the situation where contract manufacturers provide manufacturing services to brand manufacturers and sell products on third-party e-commerce platforms, the study analyzes the optimal sales strategy combinations for brand manufacturers and contract manufacturers. Specifically, this study aims to investigate the following issues under the consideration of differences in product service quality levels: (1) the invasion strategy of contract manufacturers (i.e., whether to engage in invasion), (2) the selection strategy of invasion sales modes by contract manufacturers, and (3) the impact of different commission rates and product service quality levels on the equilibrium sales mode combinations of brand manufacturers and contract manufacturers. Additionally, this study aims to explore the basis for sales mode selection among supply chain members influenced by differences in service quality levels. The management insights of this study can provide valuable decision guidance to relevant enterprises (both the contract manufacturer intending to introduce their own brands and the brand manufacturer seeking to maintain traditional brand competitiveness), and carry significant theoretical implications.

The rest of this paper is organized as follows. Related literature is reviewed in Section 2. A basic model is presented in Section 3, an analysis for equilibrium results in Section 4. Section 5 studies the strategy choice of contract manufacturer and brand manufacturer about equilibrium sales mode combinations using numerical simulation method. The last section concludes and points out directions for future research.

## 2. Literature review

Existing literature on the selection of Original Brand Manufacturing (OBM) invasion sales modes in contract manufacturing primarily focuses on three main aspects: sales mode; own brand; service quality.

Firstly, this study relates to the optimal sales mode among supply chain members. Currently, in the highly competitive online retail market environment, the choice of sales mode has become extremely crucial. Selecting an appropriate sales mode has become an imperative issue for online store operations. The choice of e-commerce platform sales mode has evolved into a critical component of manufacturers' strategic decision-making. Abhishek et al. [9] investigated the sales mode strategy selection problem for two e-retailers in the supply chain structure of a manufacturer, namely, when to adopt an agency sales mode instead of the traditional resale mode. The research found that the agency mode is more efficient, leading to lower retail prices; however, it also signifies that online retailer transfers pricing authority for the products to the manufacturer. Zhang and Hou [10] emphasized the key factors that manufacturers and e-commerce companies consider when selecting online sales models, indicating that preferences for agency sales or reselling depend on factors such as commission costs and brand advantage. Cheng et al. [11] explored the competitive dynamics between manufacturers and electronic retailers by examining the interaction between the retailer's own brand and green technology under different sales modes. The study found that although manufacturers and e-commerce retailers typically had opposing preferences regarding sales models, they both tended to prefer the reselling mode when the percentage fee was moderate. Wei and Dong [12] studied the interactive effects of different quality products and different sales modes and explored the issue of sales model selection for products of varying qualities. Zhang and Zhang [13] discussed the demand-sharing strategies between e-retailers and physical suppliers under the sales modes of agency sales and reselling. Xuemei and Jiajia [14] investigated the selection

of sales modes for manufacturers on e-commerce platforms under different risk preferences. Sun et al. [15] compared the profits of online direct sales, platform flagship stores, and wholesale sales channels, and explored the decision-making basis for manufacturers' sales mode selection. Qin et al. [16] examined the interaction between suppliers and e-commerce platforms regarding sales modes and logistics services, and explored the optimal combination strategy of sales modes and logistics services. It can be observed that when it comes to the choice of sales strategy, the research has mainly approached it from the perspective of manufacturers or e-commerce platforms, with limited investigation into the interactive dynamics between manufacturers and their competitive product sales strategies. However, this issue is indeed a crucial consideration for those intending to establish own brands through online channels, thus forming the primary motivation for the research in this paper. This study effectively bridges this gap in the research field.

Secondly, this study also involves relevant literature on own brands. Own brands have always been a hot topic in academic research. The literature on own brands has been explored from various perspectives, including retailers, manufacturers, and consumers [17]. Broadly, it can be categorized into decisions regarding the introduction of own brands and the competition between own brands and existing branded products in the market. This is also one of the main themes of this study. Fan and Chen [18] studied the impact of retailer own brand introduction on channel competition in a supply chain system where two manufacturer brands compete with each other, considering different channel power structures, and discovered that the introduction of store brands reduces the profits of brand manufacturers. Li et al. [19] studied the impact of introducing own brands by retailers on the profits of channel members under different channel strategies. They found that while the introduction of own brands by retailers could reduce manufacturer profits, it could increase overall channel profits. Cheng et al. [20] investigated the effects of retailer store brand (own brand) introduction on the entire supply chain system from a three-tier supply chain perspective (manufacturer-distributor-retailer). Some scholars have conducted research on the factors influencing the decision to introduce own brands from a reverse perspective. Duan et al. [21] studied the impact of platform channel introduction on own brand introduction decisions under the condition of endogenous own brand decisions in a single e-commerce platform and single manufacturer supply chain system. Jin et al. [22] studied the interaction between manufacturer channel strategies and retailers' decisions to introduce own brands. They discovered that manufacturers can effectively prevent store brand introductions by adopting a single-channel sales strategy. Additionally, researchers have also examined the interactive effects of different manufacturer advertising strategies on retailer own brand introduction decisions from an advertising perspective [23]. Furthermore, some researchers have focused on studying the competitive dynamics between own labels and established market brands. Li et al. [24]explored the interactive mechanism between retailer own brand introduction and manufacturer direct distribution channel establishment through a game-theoretic model, revealing that both retailers and manufacturers can achieve a win-win outcomes when retailers introduce relatively lower-quality own label products. On the other hand, Choi and Fredj [25] studied the competition between the manufacturer's national brand and the own labels of two retailers. They found that competitive retailers should focus on differentiating their private label products from those of their competitors while reducing differentiation from well-established national brands. In summary, previous researches have extensively examined own brand from various perspectives. However, much of these researches has predominantly focused downstream members of the supply chain introducing their own brands such as the e-commerce platform. This study, conversely, addresses a gap in the literature by considering the unique dynamics of contract manufacturers introducing their own brands into online channels. This offers valuable insights for

a deeper exploration of the relationships among different participants in the supply chain about own brands and their sales mode decisions.

Lastly, with intensifying market competition and increasingly diverse consumer demands, many retail enterprises are taking measures to further consolidate and expand their market presence. These measures include diversifying product offerings, enhancing user experiences, and catering to personalized demands. The advent of the service economy era has made service quality a core competitive factor in business development. Enhancing service quality has become a crucial issue that must address on the path to success in e-commerce operations. The increasing significance of service quality as an important factor in product competition has gained considerable scholarly attention. Some scholars have studied the impact of service levels on various aspects such as product prices, demand, and channel strategies. Han et al. [26] explored the impact of product service quality on wholesale and retail prices within a mixed operation model. Dan et al. [27] investigated the influence of retail service quality and customer loyalty on pricing decisions for manufacturers and retailers within a dual-channel framework. Zhang et al. [28] explored how service quality affects the decision of e-platform to establish self-operated channels when considering the impact of channel demand under varying levels of service quality and customer sensitivity. Wang et al. [29] investigated the effects of service providers improving service quality on the interests of various parties within a product service supply chain, emphasizing the significance of service quality to supply chain members and social welfare. Chen et al. [30] discovered that enhancing retail service quality can boost market demand for products under conditions of product quality homogeneity. Additionally, some researchers, similar to this study, treat service quality as a background condition and investigate operational strategy issues within a service-oriented supply chain context, including channel selection and sales mode decisions, etc. Chen et al. [31]conducted research on the optimal dual-channel strategy for manufacturers facing varying levels of service quality. Yang et al. [32] studied manufacturers' service provision strategies for two competing retailers and found that providing uniform services to both retailers may not always yield favorable outcomes. Tian et al. [33] discussed the choice between resale, market, and hybrid sales modes under conditions where different supply chain members bear service costs. Liu and Chen [4] investigated the impact of data resources and sharing strategies, as sources of sustainable competitive advantage, on pricing strategies within a product service supply chain. Xia et al. [34] investigated the reasons behind the discrepancy between manufacturers' expected service levels for enhancing product impact commitment and the actual service levels provided. Moreover, some scholars have extensively examined the operational model selection concerning service quality from multiple dimensions [35, 36]. It is evident that while there is a multitude of existing research on service quality, there has been limited consideration for online product service quality. Given that online channels, as channels where product value cannot be directly perceived, underline the importance of service quality, this study fills a research gap and offers a new perspective form service quality and enriched literature foundation for understanding the selection of online channel sales modes.

Closely related to this study are the articles of Shi [7] and Chen et al. [37], which also address the question of whether contract manufacturers should establish their own brands and engage in factory encroachment. However, the distinction lies in the fact that these articles exclusively examine the intrusion decisions of contract manufacturers without considering the scenario of online channel intrusion, thereby lacking consideration for sales modes and service quality which are important factors and ought to be considered.

Based on the review of existing literature, it can be observed that current research primarily focuses on the dual-channel strategies or sales mode selection issues between manufacturers and retailers, or between platform retailers and retailers within a two-tier supply chain. Few

studies have explored the sales mode selection problem within a three-tier supply chain system involving upstream contract manufacturers, brand manufacturers, and third-party electronic platform retailers. Regarding the literature on own brands, most studies have considered the own brands of electronic retailers, with limited research on modeling and analyzing the decision process for contract manufacturers in introducing own brands. In the literature about service quality, there is currently a dearth of research on online product service quality, a crucial aspect often overlooked, especially in the context of online consumption. This study, by examining the introduction strategy of contract manufacturers' own brands, investigates the optimal sales mode combinations for both brand manufacturers and contract manufacturers. This research aims to fill the gap in the existing literature in this domain and enrich the body of research on sales mode strategy choices within the platform supply chain structure in the e-commerce field.

## 3. Model

Consider a supply chain structure containing a contract manufacturer (i.e., CM, contract manufacturer), brand manufacturer (i.e, BM, brand manufacturer) and a third-party e-commerce platform retailer (e-R, e-retailer). Here we represent CM, BM, and e-R, with C, B, E respectively. Following the practice of multiple e-commerce platforms (such as Amazon, Jingdong and Taobao), brand manufacturers entrust contract manufacturers to produce their products at a contract price $w$ and sell the product NB (national brand) through third-party e-commerce platforms. There are two e-commerce platforms' sales modes, namely resale (Reselling) and agency sales mode (Agency selling). In the reselling mode, the retail pricing of the product is determined by the e-commerce platform, while in the agency mode, the retail pricing is determined by the manufacturer. Thus, there are two supply chain scenarios: Scenario A, where the brand manufacturer sells product NB through the agency mode, and Scenario R, where the brand manufacturer sells product NB through the reselling mode. Additionally, due to the accumulated technical experience in supplying BM, CM has the capacity to circumvent the brand manufacturer and engage in the production of competitive products, namely the product SB (store brand, i.e., own brand), for the manufacturing of product NB. Subsequently, CM can sell product SB through third-party e-commerce platforms. Both types of products are accompanied by their respective services and these services affect the corresponding market demand. Both the brand and the contract manufacturer are free to choose the sales mode for their own brands. Therefore, if the contract manufacturer chooses brand intrusion, there are four possible supply chain structures, as shown in Fig 1.

Based on the aforementioned supply chain models, this study discusses the invasion decision and the selection of invasion sales modes by the contract manufacturer considering the presence of disparities in product service quality levels.

### 3.1 Demand function

Table 1 lists the notations used in this study and their definitions. Refer to the literature by Tian, Vakharia, Tan and Xu [33], In this study, we denote the market demand for the brand manufacturer's product and the contract manufacturer's product as $D_i$, $i = N$, $S$, respectively: $D_S = \theta + \lambda s_S - p_S + \gamma(p_N - p_S)$ means the demand for brand manufacturer's product $SB$ and $D_N = \theta + \lambda s_N - p_N + \gamma(p_S - p_N)$ for the product NB, and $\theta$ indicates the market potential in this study. To specifically examine the influence of product service quality disparities on the contract manufacturers' own-brand invasion and sales mode selection, it is assumed that the market potential for both the contract manufacturers' product and the brand manufacturers' product is equal. This assumption, while not impacting the ultimate findings, enables a

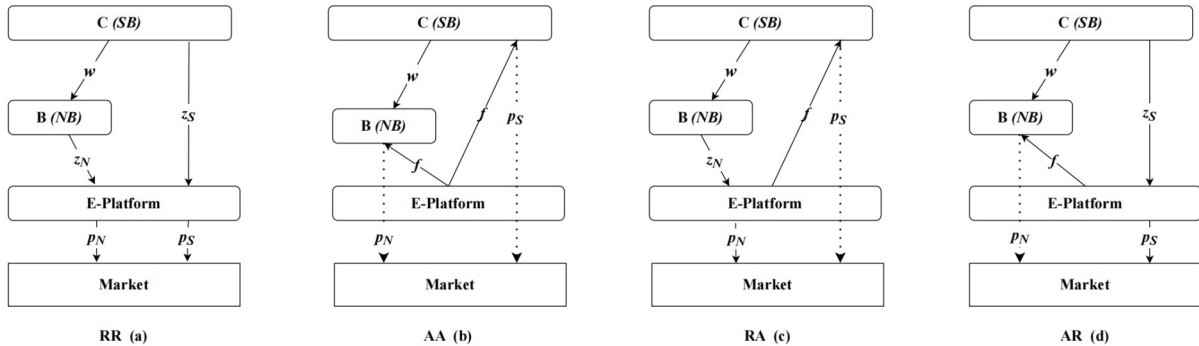

**Fig 1. Supply chain structure of four different sales mode combination.** The four supply chain scenarios are as follows: (1) Scenario RR: Both the brand manufacturer and the contract manufacturer opt for the reselling sales mode, as illustrated in Fig 1(a); (2) Scenario AA: Both the brand manufacturer and the contract manufacturer choose the agency sales mode, as illustrated in Fig 1(b); (3) Scenario RA: The brand manufacturer selects the reselling sales mode, while the contract manufacturer adopts the agency sales mode, as shown in Fig 1(c); (4) Scenario AR: The brand manufacturer chooses the agency sales mode, while the contract manufacturer goes for the reselling sales mode, as presented in Fig 1(d).

comprehensive elucidation of the underlying mechanism by which quality levels impact the selection of sales modes. $p_N$ and $p_S$ denote the market prices of the respective products. $s_i > 0$, $i = N, S$, $s_S$ and $s_N$ represent the product service quality levels of the contract manufacturer and the brand manufacturer, respectively, with a higher value of $s_i$ indicating a higher product service quality level. It is worth noting that $s_i$ represents the relative service quality level, thus $s_i > 0$. Additionally, it with its sensitivity coefficient $\lambda$, causes joint effect to influence product demand, allowing changes in the service level parameter to have a significant impact on product demand. Without loss of generality, the service quality cost is denoted as $ks_i^2$ in this study, and it is normalized for ease of analysis, with the service cost represented as $s_i^2$.

**Table 1. Notations.**

| notation | definition |
|---|---|
| $\theta$ | The initial market basic demand, this study assumes the same for both products |
| $f$ | Exogenous commission rate |
| $\lambda$ | Sensitivity coefficient of service quality level |
| $s_N$ | Service quality level of the brand manufacturer's products |
| $s_S$ | Service quality level of the contract manufacturer's products |
| $\gamma$ | Price elasticity coefficient, reflecting consumer sensitivity to price and product competitiveness |
| $p_N$ | Price of the product NB |
| $p_S$ | Price of the product SB |
| $D_N$ | Market demand for the product NB |
| $D_S$ | Market demand for the product SB |
| $w$ | Wholesale price, i.e., the price at which the brand manufacturer purchases from the contract manufacturer |
| $z_N$ | Resale price of product NB, i.e., the price at which the brand manufacturer resells to the e-commerce platform |
| $z_S$ | Resale price of product SB, i.e., the price at which the contract manufacturer resells to the e-commerce platform |
| $\pi_C$ | Profit of the contract manufacturer |
| $\pi_B$ | Profit of the brand manufacturer |
| $\pi_E$ | Profits of e-commerce platform (e-Retailer) |

Additionally, for analysis convenience, it is assumed that the production cost of the products is standardized to 0. $\lambda$ represents the sensitivity of market demand to service quality, where a larger value of $\lambda$ indicates higher consumer sensitivity to service quality. Furthermore, in accordance with established literature [9, 10, 16], and without loss of generality, it is assumed that the commission rate $f$ is exogenous. The e-commerce platform in the model is a hybrid sales platform and does not exhibit a particular preference for any specific sales mode.

### 3.2 Sequence of the events

As this study primarily focuses on the decision-making of contract manufacturers regarding their invasion sales modes, we initially assume that the sales mode decision of the brand manufacturer is exogenously fixed prior to the invasion by the contract manufacturer. Specifically, the brand manufacturer's sales mode is assumed to be either agency mode or resale mode. In Section 4 Analysis, we will further discuss the equilibrium sales mode combination among the supply chain members while considering the endogeneity of the brand manufacturer's sales mode decision.

Fig 2 depicts the sequence of events in this game, and the sequence of decisions is motivated by some industry practices. Since the choice of sales mode by the contract manufacturer is typically a long-term decision in practice, it is placed at the beginning stage of the game. In Stage 1, the contract manufacturer decides whether to establish its own brand and which sales mode to adopt (if it chooses to establish a brand). Thus, there are a total of six scenarios, including two benchmark scenarios (without invasion) and pairwise combinations of the two sales modes, as illustrated in Stage 1 of Fig 2. The major difference between the two sales modes lies in the pricing authority: under the resale mode, pricing is determined by the third-party e-retailer, while under the agency sales mode, pricing is determined by the brand manufacturer or the contract manufacturer. This also determines the game sequence for the pricing decision

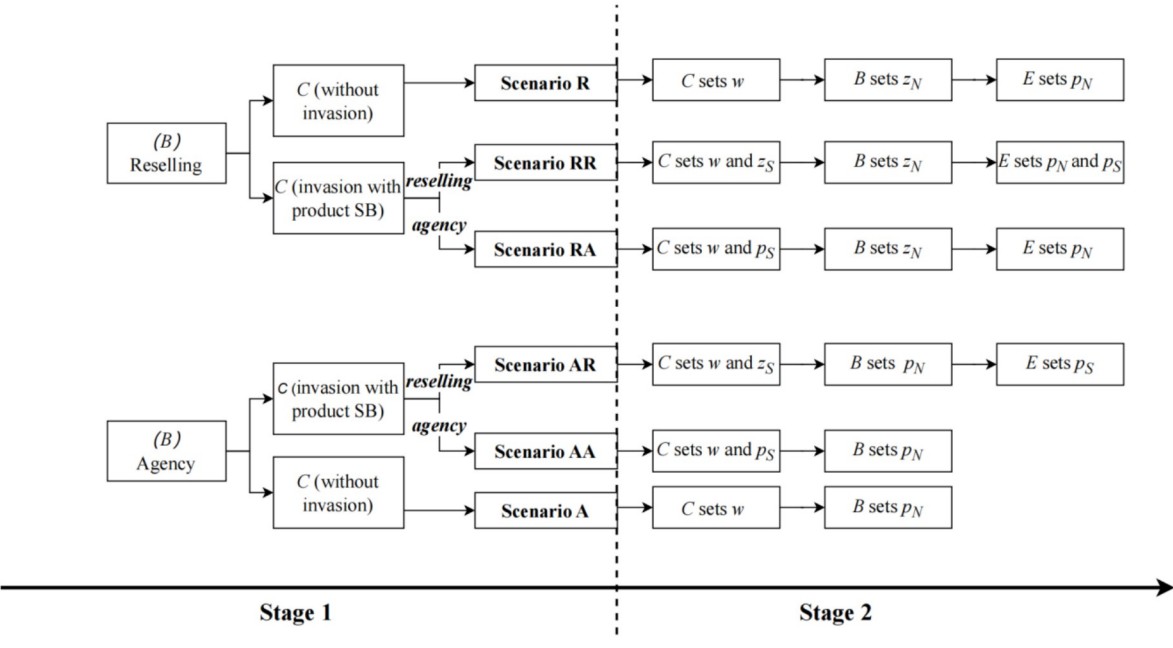

**Fig 2. Sequence of the events.**

in the second stage, as depicted in Stage 2 of Fig 2. The game sequence for each scenario will be further explained in the following sections.

### 3.3 Model establishment and solution

In this section, we firstly establish and solve the benchmark scenarios R and A, which do not involve the OBM product SB prior to invasion. The benchmark scenarios are constructed based on the reselling and agency sales modes, respectively. Then, we extend the benchmark models to the post-invasion scenarios RR, AR, RA, and AA, which include the OBM product SB.

**3.3.1 Benchmark scenario R.**   In the scenario R, it is assumed that the contract manufacturer (CM), chooses to manufacture the product NB only for the brand manufacturer (BM), but does not establish its own brand SB. Moreover, the BM adopts the reselling mode for product sales. In this scenario, the CM determines the wholesale price $w$, followed by the BM's decision on the reselling price $z_N$, and ultimately, the e-retailer determines the retail price $p_N$. Consequently, the profit distribution among the supply chain members in scenario R can be summarized as follows:

$$\begin{cases} \pi_C = wD_N \\ \pi_B = (z_N - w)D_N - s_N^2 \\ \pi_E = (p_N - z_N)D_N \end{cases} \tag{1}$$

Lemma 1: When the contract manufacturer (CM) chooses not to invade and the brand manufacturer (BM) chooses the reselling mode, the optimal decisions for the firms are as follows: $w = \frac{\theta+s_N\lambda}{2}$, $z_N = \frac{3(\theta+s_N\lambda)}{4}$ and $p_N = \frac{7(\theta+s_N\lambda)}{8}$. At the same time, the maximum profits are as follows: $\pi_C = \frac{(\theta+s_N\lambda)^2}{16}$ and $\pi_B = \frac{(\theta+s_N\lambda)^2 - 32s_N^2}{32}$.

**3.3.2 Benchmark scenario A.**   Similarly, when the contract manufacturer chooses to only manufacture product NB for the brand manufacturer and does not manufacture product SB, and the brand manufacturer chooses the agency sales mode to sell the product. The contract manufacturer determines the wholesale price $w$, followed by the brand manufacturer determining the retail price $p_N$. In this scenario, the profits of the supply chain members are as follows:

$$\begin{cases} \pi_C = wD_N \\ \pi_B = (1-f)p_ND_N - wD_N - s_N^2 \\ \pi_E = fp_ND_N \end{cases} \tag{2}$$

Lemma 2: When the contract manufacturer chooses not to invade and the brand manufacturer chooses the agency mode, the optimal decisions for the enterprises are as follows: $w = \frac{(1-f)(\theta+s_N\lambda)}{2}$ and $p_N = \frac{3(\theta+s_N\lambda)}{4}$. Similarly, the maximum profits are as follows: $\pi_C = \frac{1}{8(1-f)(\theta+s_N\lambda)^2}$ 和 $\pi_B = \frac{1}{6(1-f)(\theta+s_N\lambda)^2 - 96s_N^2}$.

**3.3.3 Scenario RR.**   Firstly, this study analyzes the scenario where both products NB and SB choose reselling mode, with the supply chain structure depicted in Fig 1(a). In this scenario, the contract manufacturer determines the wholesale price $w$, and then the brand manufacturer and the contract manufacturer independently decide the reselling prices $z_N$ and $z_S$ for NB and SB. Finally, the e-commerce platform determines the retail prices $p_N$ and $p_S$ for NB and SB, as shown in Fig 2, Stage 2, scenario RR. Combining Eqs (1) to (2), the profits of each supply

chain member are as follows:

$$\begin{cases} \pi_C = z_S D_S + w D_N - s_S^2 \\ \pi_B = (z_N - w) D_N - s_N^2 \\ \pi_E = (p_N - z_N) D_N + (p_S - z_S) D_S \end{cases} \tag{3}$$

Lemma 3: When the brand manufacturer adopts the reselling mode and the contract manufacturer chooses to invade with the reselling mode, the optimal decisions for the firms are as follows: $w = \frac{\theta(1+2\gamma) + \lambda(s_N + (s_N + s_S)\gamma)}{2+4\gamma}$, $z_S = \frac{\theta(1+2\gamma) + \lambda(s_S + (s_N + s_S)\gamma)}{2+4\gamma}$, $z_N = \frac{3s_N\lambda + 2\lambda\gamma(3s_N + s_S + (s_N + s_S)\gamma) + \theta(3 + 4\gamma(2+\gamma))}{4(1+\gamma)(1+2\gamma)}$, $p_S = \frac{3(\theta(1+2\gamma) + \lambda(s_S + (s_N + s_S)\gamma))}{4+8\gamma}$ and $p_N = \frac{1}{8}\left(6\theta + 3(s_N + s_S)\lambda + \frac{\theta + s_N\lambda}{1+\gamma} + \frac{3(s_N - s_S)\lambda}{1+2\gamma}\right)$. Furthermore, the

maximum profits for each firm are as follows: $\pi_C =$

$$\left\{ \frac{\begin{array}{c} 2s_N\theta\lambda(1+2\gamma)^2 + \theta^2(1+2\gamma)(3+4\gamma) + 4s_S\lambda(1+\gamma)(\theta(1+2\gamma) + s_N\lambda\gamma) \\ +\lambda^2 s_N^2 + (1+\gamma)(2\gamma\lambda^2 s_N^2 + 2(\lambda^2 - 8 + \gamma(\lambda^2 - 16))s_S^2) \end{array}}{16(1+\gamma)(1+2\gamma)} \right\}$$ and $\pi_B = \frac{(\theta + s_N\lambda)^2}{32(1+\gamma)} - s_N^2$.

**3.3.4 Scenario AR.** In Scenario AR, the brand manufacturer adopts the agency sales mode for product NB, while the contract manufacturer C chooses the reselling mode for product SB. Similarly, the contract manufacturer first determines the wholesale price $w$ and reselling price $z_S$, followed by the brand manufacturer deciding the retail price $p_N$ for product NB, and almost simultaneously the e-commerce platform determines the retail price $p_S$ for product SB. In this scenario, the profits of the supply chain members are as follows:

$$\begin{cases} \pi_C = z_S D_S + w D_N - s_S^2 \\ \pi_B = ((1-f)p_N - w) D_N - s_N^2 \\ \pi_E = f p_N D_N + (p_S - z_S) D_S \end{cases} \tag{4}$$

Lemma 4: When the brand manufacturer adopts the agency mode and the contract manufacturer chooses to invade with the reselling mode, the optimal decisions for each supply chain member are as follows: $w = \frac{(1-f)(\theta(1+2\gamma) + \lambda(s_N + (s_N + s_S)\gamma))}{2+4\gamma}$, $z_S = \frac{s_S\lambda - \theta(1+2\gamma)((f-1)\gamma - 1) + \lambda\gamma(2s_S - (f-1)(s_N + (s_N + s_S)\gamma))}{2(1+\gamma)(1+2\gamma)}$

$p_S = \frac{1}{8}\left(4\theta + 2(s_N + s_S)\lambda + \frac{\theta + s_S\lambda}{1+\gamma} + \frac{2(s_S - s_N)\lambda}{1+2\gamma} - \frac{2(\theta + (2+f)\theta\gamma + \lambda(s_S + (s_N + f s_N + s_S)\gamma))}{\gamma((f-1)\gamma - 4) - 2}\right)$ and

$p_N = \frac{\left(2\theta + s_N\lambda + s_S\lambda + \frac{(s_N - s_S)\lambda}{1+2\gamma} + \frac{s_S\lambda\gamma + 2s_N\lambda(1+\gamma) + \theta(2+3\gamma)}{2+\gamma(4+\gamma - f\gamma)}\right)}{4}$. At the same time, the maximum profits are as follows: $\pi_C = \pi_C^*$ and $\pi_B = \frac{\left\{ \begin{array}{c} 4(f-1)s_N\lambda(1+\gamma)(s_S\lambda\gamma + \theta(2+3\gamma)) + (f-1)(s_S\lambda\gamma + \theta(2+3\gamma))^2 \\ +4s_N^2(1+\gamma)(16 + (f-1)\lambda^2(1+\gamma) + 8\gamma(4+\gamma - f\gamma)) \end{array} \right\}}{32(1+\gamma)(\gamma((f-1)\gamma - 4) - 2)}$.

Due to the complexity of the equilibrium expression for variable $\pi_C$ in Scenario AR, we substitute it with variable $\pi_C^*$ in Scenario AR omit the results of its specific expression. The specific expression for variable $\pi_C^*$ can be found in the S1 Appendix proof of lemma 4.

**3.3.5 Scenario RA.** In Scenario RA, the contract manufacturer first determines the wholesale price $w$, followed by the immediate decision of the retail price $p_S$. Then, the brand manufacturer decides on the resale price $z_N$, and subsequently, the e-commerce platform determines the retail price $p_N$ of product NB. Similarly, the profits of the supply chain members in this scenario are as follows:

$$\begin{cases} \pi_C = (1-f)p_S D_S + w D_N - s_S^2 \\ \pi_B = (z_N - w) D_N - s_N^2 \\ \pi_E = (p_N - z_N) D_N + f p_S D_S \end{cases} \tag{5}$$

Lemma 5: When the brand manufacturer adopts the reselling mode and the contract manufacturer chooses to invade with the agency mode, the optimal decisions of the supply chain members are as follows: $w = \frac{\theta(1+2\gamma)(1+(1-f)\gamma)+\lambda(s_N(1+2\gamma)+(1-f)(\gamma^2(s_S+s_N)+s_S\gamma))}{2(1+\gamma)(1+2\gamma)}$,

$z_N = \frac{3s_N\lambda}{4(1+\gamma)(1+2\gamma)} - \frac{\theta(2(f-1)\gamma-3)+2\lambda\gamma(3s_N+(1-f)((s_N+s_S)\gamma+s_S)}{4(1+\gamma)}$, $p_S = \frac{\theta(1+2\gamma)+\lambda(s_S+(s_N+s_S)\gamma)}{2+4\gamma}$ and

$p_N = \frac{(1+\gamma)(1+2\gamma)}{8((1+2\gamma)(7+4\gamma)\theta+(7+2\gamma(7+2\gamma))\lambda s_N+4\gamma(1+\gamma)\lambda s_S)}$. Similarly, the maximum profits for each member of

the supply chain are as follows: $\pi_C = \frac{\left\{ \begin{array}{c} \theta^2(1+2\gamma)(4f+8(f-1)\gamma-5)+4s_S^2(1+\gamma)(4+8\gamma+(f-1)\lambda^2(1+\gamma))+ \\ 8(f-1)s_S\lambda(1+\gamma)(\theta(1+2\gamma)+s_N\lambda\gamma)+s_N^2\lambda^2(4(f-1)\gamma^2-1-2\gamma)+2s_N\theta\lambda(1+2\gamma)(4(f-1)\gamma-1) \end{array} \right\}}{-16(1+\gamma)(1+2\gamma)}$

and $\pi_B = \frac{1}{8(\theta+s_N\lambda)}$.

**3.3.6 Scenario AA.** Similar to scenario RA, in scenario AA, the contract manufacturer first determines the wholesale price $w$, followed by the brand manufacturer and contract manufacturer independently determining the retail prices $p_N$ and $p_S$ for products NB and SB. Hence, the respective profits of each supply chain member in this scenario can be expressed as follows:

$$\begin{cases} \pi_C = (1-f)p_S D_S + wD_N - s_S^2 \\ \pi_B = ((1-f)p_N - w)D_N - s_N^2 \\ \pi_E = f(p_S D_S + p_N D_N) \end{cases} \tag{6}$$

Lemma 6: When the brand manufacturer adopts the agency mode and the contract manufacturer chooses to invade with the agency mode, the optimal decisions for supply chain members are as follows: $w = \frac{(1-f)(\theta+2\theta\gamma+\lambda(s_N+(s_N+s_S)\gamma))}{2+4\gamma}$, $p_N = \frac{3s_N\lambda+2\lambda\gamma(3s_N+s_S+(s_N+s_S)\gamma)+\theta(3+4\gamma(2+\gamma))}{4(1+\gamma)(1+2\gamma)}$ and

$p_S = \frac{\theta+2\theta\gamma+\lambda(s_S+(s_N+s_S)\gamma)}{2+4\gamma}$. Furthermore, the maximum profits for each supply chain member in

this scenario are as follows: $\pi_C = \frac{(1-f)\left( \begin{array}{c} (1+2\gamma)(3+4\gamma)\theta^2+2(1+2\gamma)^2\theta\lambda s_N+(1+2\gamma(1+\gamma))\lambda^2 s_N^2 \\ +4s_S(\gamma\lambda s_N+\theta(1+2\gamma))\lambda(1+\gamma)+2\lambda^2 s_S^2(1+\gamma)^2 \end{array} \right)}{8(1+\gamma)(1+2\gamma)} - s_S^2$ and

$\pi_B = \frac{(1-f)\theta^2-s_N\left( 2(f-1)\theta\lambda+(f-1)s_N\lambda^2+16s_N(1+\gamma) \right)}{16(1+\gamma)}$.

## 4. Analysis

In this section, based on the equilibrium outcomes of the supply chain under the benchmark scenarios and incorporating the equilibrium outcomes after the invasion, an analysis and comparison of profits and demand under different scenarios are conducted. The analysis aims to examine the factors influencing the choice of sales mode in the invasion decision of the contract manufacturer and derive corresponding propositions.

**Proposition 1:** Both for the brand manufacturer and the contract manufacturer, the demand for products under the agency mode is always greater than the demand for products under the resale mode.

This conclusion is consistent with the findings of many previous studies, such as Cai et al. [38]. It can be inferred that the reason for this difference is that, under the agency mode, where the pricing authority lies with the brand manufacturer or the contract manufacturer, the supply chain structure is shorter and there is no double marginalization effect. As a result, prices can be lower, leading to higher product demand, which better reflects the actual market demand without deadweight loss. It is worth noting that for the brand manufacturer, in both scenario RA and scenario RR, the market demand for the brand manufacturer's product remains the same. This finding is also consistent with the following Proposition 3.

**Proposition 2:** When the brand adopts the reselling mode, the contract manufacturer chooses to invade if and only if $\theta > \hat{\theta}^R$, ($\hat{\theta}^R = \max\{\theta_1, \theta_2\}$). When the brand adopts the

agency mode, the contract manufacturer chooses to invade if and only if $\theta > \hat{\theta}^A$, ($\hat{\theta}^A = \max\{\theta_3, \theta_4\}$).

According to Proposition 2, regardless of whether the brand adopts the agency mode or the reselling mode, the contract manufacturer will choose to invade as long as the initial market demand is sufficiently large. This can be easily understood as a larger market space can accommodate more product competition. As shown in Fig 3 ($s_N = 0.5$; $s_S = 0.4$; $\lambda = 0.8$; $\gamma = 0.6$) illustrating the impact of commission rates on the threshold of $\theta$ when it is used as a criterion for determining whether contract manufacturers invade, when is $\theta$ greater than 1, the contract manufacturer can establish its own brand and obtain higher profits. This partly explains why many contract manufacturing companies that initially focused on contract manufacturing business eventually establish their own brands and expand their OBM business in the later stages of development. Furthermore, from Fig 3, it can be observed that the invasion threshold $\theta$ of the contract manufacturer under both sales modes of the brand manufacturer increase with the commission rate $f$. This is understandable because a higher commission rate leads to a greater profit sharing with the e-commerce platform, reducing the additional profits of contract manufacturer gained from brand establishment and invasion under the agency mode. As a result, a larger market demand is required, ultimately leading to an increase in the invasion threshold. It is worth noting that at high commission rates, the invasion threshold under the agency mode $\hat{\theta}^A$ is lower than the threshold under the reselling mode $\hat{\theta}^R$. This is because at high commission rates, denoted by $\hat{\theta}^R$ and $\hat{\theta}^A$ respectively, refer to $\theta_2$ (invasion threshold in the RA scenario) and $\theta_4$ (invasion threshold in the AA scenario). The rate of decline in the profit difference between the RA scenario and the R scenario is higher as $f$ decreases, compared to the rate of decline in the profit difference between the AA scenario and the A scenario. Therefore, the invasion threshold can be reached faster with commission rate $f$ increasing when brand manufacturer chooses agency mode. While the profit of the contract manufacturer in scenario A also decreases with the increase in the commission rate $f$, the profit of the contract manufacturer in scenario R remains unaffected by the commission rate. Thus, at the same market demand threshold $\theta$, the corresponding commission rate for the reselling mode is smaller than that for the agency mode. In other words, when the commission rates are high and equal, the invasion threshold $\hat{\theta}^A$ for agency mode is smaller than the threshold $\hat{\theta}^R$ for the reselling mode.

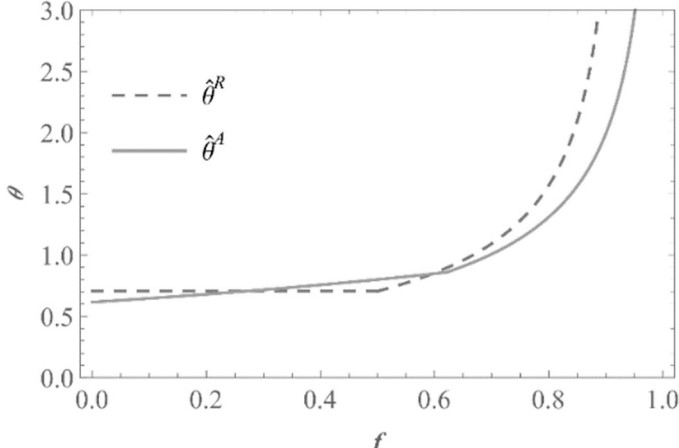

**Fig 3. Effect of commission rate $f$ on invasion threshold $\theta$ ($s_N = 0.5$; $s_S = 0.4$; $\lambda = 0.8$; $\gamma = 0.6$).**

It is worth noting that for contract manufacturers intending to engage in brand creation, the potential demand for their own brands can be estimated through the demand (or profitability) of the brand manufacturer's products they contract for. If the profitability of the brand manufacturer's products they are contracting for is high, it indicates a large potential demand for the product, making brand building invasion a profitable choice in this situation.

**Proposition 3:** (1) When $s_N$, $s_S$, $\gamma$, $\lambda \in (0, 1)$ and $\theta > 1$, the threshold $\bar{f}^A$ for the contract manufacturer to choose the agency/reselling mode of invasion under the situation brand manufacturer choosing agency mode is greater than the threshold $\bar{f}^R$ for the contract manufacturer to choose the agency/reselling mode under the situation brand manufacturer choosing reselling mode.

(2). (i) When $f < \bar{f}^R$, regardless of whether the brand manufacturer adopts the reselling or agency mode, the contract manufacturer will choose the agency mode to invade.

(ii) When $\bar{f}^R < f < \bar{f}^A$, if the brand manufacturer adopts the reselling mode, the contract manufacturer will choose the reselling mode to invade; if the brand manufacturer adopts the agency mode, the contract manufacturer will choose the agency mode to invade.

(iii) When $f > \bar{f}^A$, regardless of whether the brand manufacturer adopts the reselling or agency mode, the contract manufacturer will choose the reselling mode of invasion, where $\bar{f}^A$ is the solution of $\pi_C^{AR} - \pi_C^{AA} = 0$ in the interval $(0, 1)$.

The threshold of $f$ (i.e., $\bar{f}$) objectively represents the critical point at which contract manufacturers choose their sales mode. $\bar{f}^R$ represents the threshold for the contract manufacturer to choose between the agency and reselling modes when it intrudes under the situation brand manufacturer choosing reselling mode. In other words, commission rate $f < \bar{f}^R$ indicates the contract manufacturer opting for the agency mode to invade, otherwise selecting the reselling mode. Similarly, $\bar{f}^A$ signifies the threshold for the contract manufacturer to decide between the agency and reselling modes when intruding under the situation brand manufacturer choosing agency mode.

Proposition 3 (1) suggests that under the brand manufacturer's agency mode, the contract manufacturer is more likely to choose the agency mode over the reselling mode. In terms of decision-making for the contract manufacturer, the brand manufacturer's agency mode promotes the contract manufacturer to also opt for the agency mode, resulting in competition within the same market. This supply chain structure encourages competition between the brand manufacturer and the contract manufacturer. The difference in profits between the contract manufacturer's choice of agency mode and reselling mode under the brand manufacturer's reselling mode changes at a higher rate with the increase of $f$ compared to the difference in profits between the contract manufacturer under the brand manufacturer's agency mode with the change in $f$. That's to say, in this scenario, the decrease in profits due to double marginalization is greater than the decrease in profits caused by the increase in commission rate $f$. As a result, the contract manufacturer is more inclined to choose the agency mode, despite the increase in the commission rate.

It is worth noting that in the reselling mode of the brand owner, it can be observed that the value of $\pi_C^{RR} - \pi_C^{RA}$ is only related to the exogenous commission rate parameter $f$. In other words, the threshold $\bar{f}^R$ is constant and independent of other model parameters such as the exogenous commission rate, and the threshold value $\bar{f}^R$ forms a straight line. As shown in Figs 4 & 5, it displays the criteria for the selection of sales modes under the brand manufacturer reselling mode or agency selling mode, where the commission rate threshold $\bar{f}$ as a standard for sales mode selection varies with the product service quality level of either the brand manufacturer or the contract manufacturer. To put it another way, the choice of sales mode for the

contract manufacturer's invasion is only dependent on the exogenous commission rate $f$ and is not influenced by the service quality level of the brand manufacturer's or contract manufacturer's products. In this case, when the brand manufacturer adopts the reselling mode, the pricing authority belongs to the third-party e-commerce platform, which becomes a direct competitor to the contract manufacturer. The variation in the quality level of the brand manufacturer's products, due to the presence of double marginalization, cannot directly affect the final product NB retail price and thus influence the market competition. In this situation, the brand manufacturer operates in a monopolistic market where the e-commerce platform functions as the ultimate consumer. The absence of pricing authority for the brand manufacturer means that it cannot participate in the competition with the contract manufacturer. Therefore, the choice of sales mode for the contract manufacturer, which is based on the threshold value of the commission rate $f$, remains constant regardless of the competition advantages or disadvantages resulting from the variation in product service quality level and cannot be reflected in the final retail prices of the products.

Similarly, akin to the scenario when brand manufacturer opts reselling mode, the contract manufacturer opts for reselling mode to invade at lower commission rates, whereas they choose agency mode to invade at higher commission rates. However, what sets this apart is that the commission rate threshold $\bar{f}^A$ is not a constant in this case. Specifically, under brand manufacturer adopting agency mode, the threshold $\bar{f}^A$ for contract manufacturers to select the invasion sales mode is not only greater than $\bar{f}^R$ but also dependent on the service quality levels of the two products. More specifically, namely $\frac{\partial \bar{f}^A}{\partial s_S} < 0$ and $\frac{\partial \bar{f}^A}{\partial s_N} > 0$. As shown in Figs 4 & 5, it is evident that the threshold value $\bar{f}^A$ exhibits a positive monotonic relationship with the brand

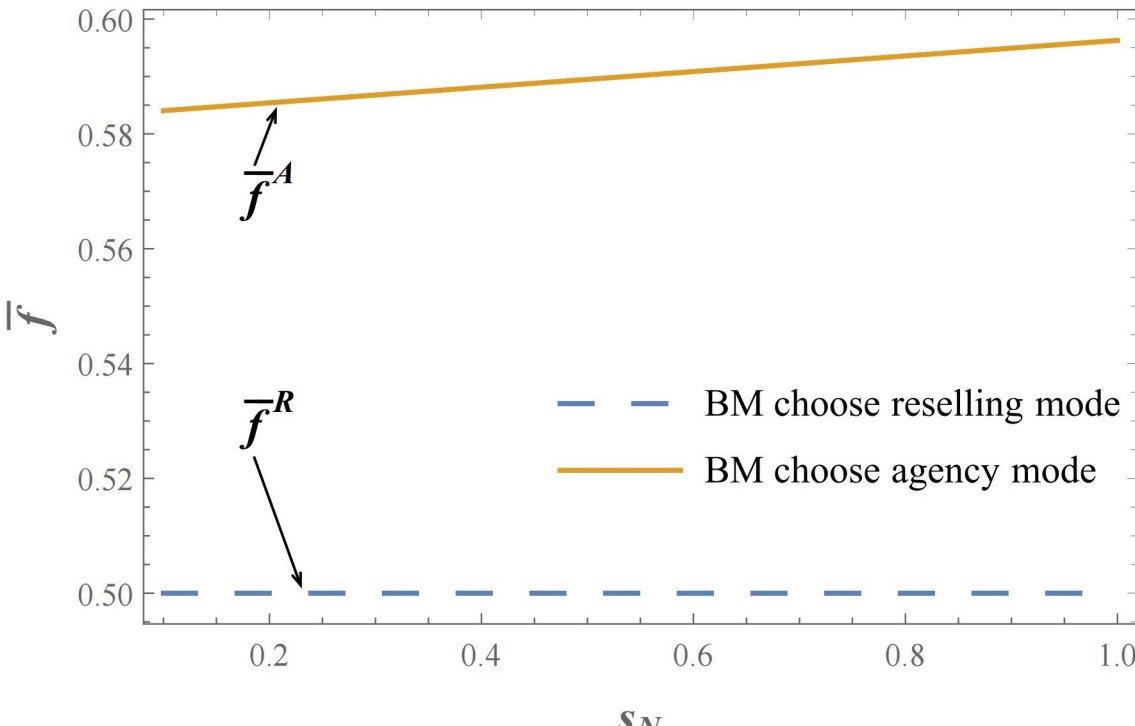

**Fig 4. Under the brand manufacturer reselling mode or agency selling mode, the influence of the product service quality level of brand manufacturer on the threshold $\bar{f}$, namely $s_N$ effect on the threshold $\bar{f}$ ($\theta = 2$; $s_S = 0.5$; $\lambda = 0.8$; $\gamma = 0.6$).**

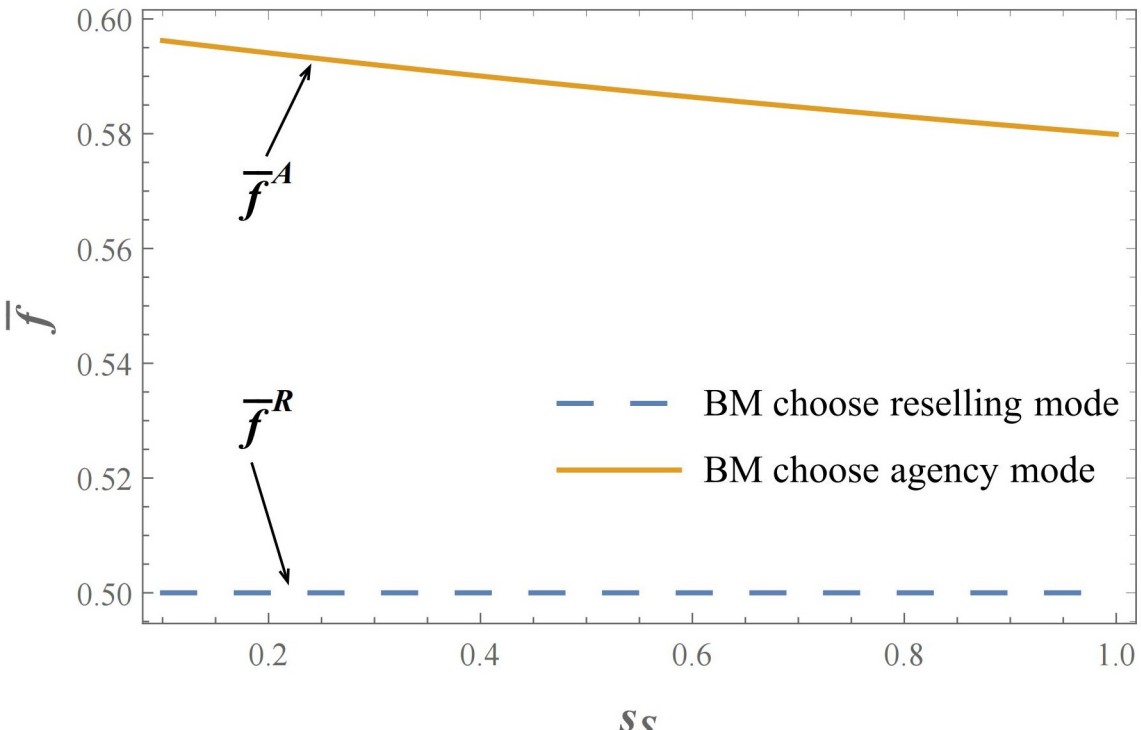

**Fig 5. Under the brand manufacturer reselling mode or agency selling mode, the influence of the product service quality level of contract manufacturer on the threshold $\bar{f}$, namely $s_S$ effect on the threshold $\bar{f}$ ($\theta = 2$; $s_N = 0.5$; $\lambda = 0.8$; $\gamma = 0.6$).**

manufacturer's product service quality level $s_N$, while demonstrating a negative monotonic relationship with the contract manufacturer's product service quality level $s_S$. Consequently, the following propositions can be derived from this observation.

**Proposition 4:** When the brand manufacturer adopts the agency mode, a higher level of brand manufacturer's service quality increases the probability of the contract manufacturer choosing the agency mode to invade. Conversely, a higher level of product service quality for the contract manufacturer increases the probability of choosing the reselling mode.

Based on the previous discussion, it is not difficult to demonstrate that $\frac{\partial \bar{f}^A}{\partial s_S} < 0$ and $\frac{\partial \bar{f}^A}{\partial s_N} > 0$, namely an increase in the service quality level of the brand manufacturer($s_N$), leading to an increase in the threshold $\bar{f}^A$, whereas an increase in the service quality level of the contract manufacturer ($s_S$), leads to a decrease in the threshold $\bar{f}^A$. When brand manufacturer opts for the agency mode, it is worth noting that an increase in the relative strength of the contract manufacturer's service quality level actually makes it more likely for the contract manufacturer to choose the reselling mode to invade (due to the decrease of threshold $\bar{f}^A$). Consequently, the contract manufacturer is more inclined to relinquish pricing authority, allowing e-commerce platforms and brand manufacturers to engage in price competition, thereby attaining greater profits compared to the agency mode. A reduction in the threshold value is observed when contract manufacturers opt for reselling mode rather than agency mode when their service quality level improves. Therefore, for the contract manufacturer, under the agency mode of the brand manufacturer, a lower service quality level makes it more suitable for the contract manufacturer to invade using their own brand products in the agency mode, while a higher

service quality level makes their own products more suitable for the reselling mode, leaving pricing and sales for the e-commerce platform. This provides us with a valuable managerial insight.

## 5. Strategy discussion

To discuss the sales mode decisions of brand manufacturers and the equilibrium sales mode combinations between contract manufacturers and brand manufacturers under different exogenous parameter conditions, this section assumes that the brand manufacturers can first determine their own sales mode, and then the contract manufacturers decide their invasion sales mode. Since the equilibrium solutions and profit functions of the aforementioned model are complex, making direct comparisons between profit functions inconvenient, this section will utilize numerical analysis to study the equilibrium sales mode combinations between brand manufacturers and contract manufacturers under different scenarios.

According to Proposition 2, when the value of $\theta$ is relatively large (i.e., $\theta > \max\left\{\hat{\theta}^R, \hat{\theta}^A\right\}$), the contract manufacturer will opt for invasion. Referring to the study conducted by Zhang, Li and Lv [28], assuming that $\theta = 2$; $\lambda = 0.8$; $\gamma = 0.6$, where $\theta$ is greater than the invasion threshold, this section will discuss the equilibrium sales mode combinations between the contract manufacturer and the brand manufacturer, considering different levels of brand manufacturer's product service quality($s_N$), varying commission rates $f$ and the product service quality of the contract manufacturer ($s_S$)following the decision to invade. As shown in Fig 6 which presents the impact of parameters $s_S$, $s_N$ and $f$ on the equilibrium sales mode combination strategy for both the brand manufacturer and the contract manufacturer, the combinations RR, RA, AR, and AA represent the sales mode combinations between the brand manufacturer and the contract manufacturer, where RR represents reselling mode for both, RA represents reselling mode for the brand manufacturer and agency mode for the contract manufacturer, AR represents agency mode for the brand manufacturer and reselling mode for the contract manufacturer, and AA represents agency mode for both. It is evident that the results of the study demonstrate a significant impact of the identified parameters.

It can be observed that regardless of the service quality levels of the brand manufacturer or the contract manufacturer, when the commission rate $f < 0.5$, the AA scenario, where both the

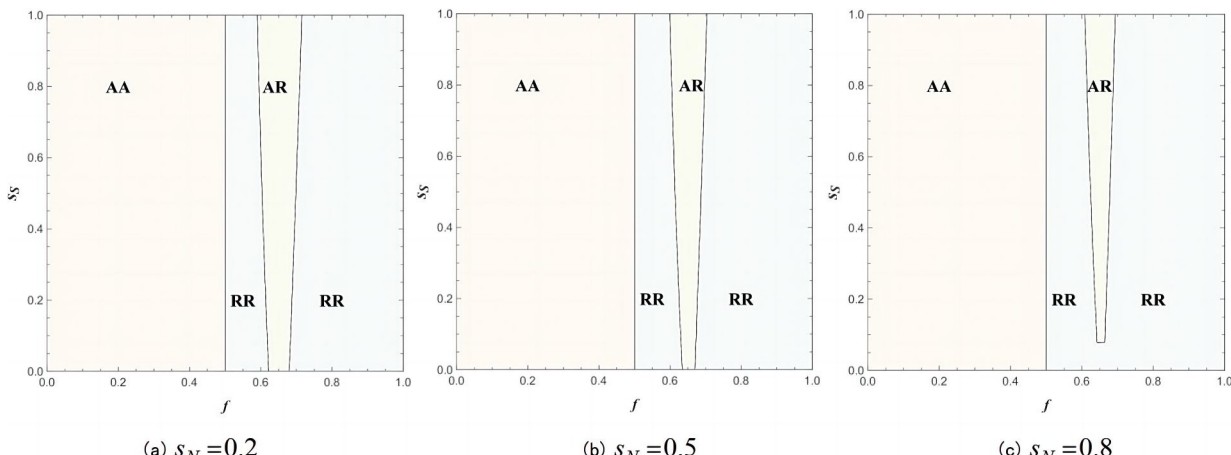

(a) $s_N = 0.2$ (b) $s_N = 0.5$ (c) $s_N = 0.8$

**Fig 6. The impact of $s_S$, $s_N$ and $f$ on the balanced sales mode, combination of brand manufacturer and contract manufacturer($\theta = 2$; $\lambda = 0.8$; $\gamma = 0.6$).**

brand manufacturer and the contract manufacturer choose the agency mode, becomes the equilibrium sales mode combination. In this case, the product service quality level does not affect the equilibrium sales mode combination. This finding is consistent with Proposition 3, as when the commission rate $f < 0.5$, the contract manufacturer will only choose the agency mode. Furthermore, the brand manufacturer's profit in the AA scenario is always greater than the brand manufacturer's profit in the RA scenario, regardless of the value of $s_N$ or $s_S$, which is why the AA scenario is located in the left half of the entire graph.

Similarly, when the commission rate $f > 0.5$, the contract manufacturer will only choose the reselling mode. However, when commission rate $f$ is close to the value of threshold $\bar{f}^A$, the equilibrium sales mode combination shifts from RR to AR, and the AR space expands with an increase in the contract manufacturer's service quality level $s_S$. As shown in Fig 6, the brand manufacturer has more room to choose the agency mode in this case. This means that even though an increase in $f$ leads to a decrease in the brand manufacturer's profit in the agency mode, the increase in $s_S$ results in an overall increase in the brand manufacturer's profit. Consequently, the profit in the agency mode becomes greater than the profit in the reselling mode for the brand manufacturer. Furthermore, when comparing the equilibrium outcomes for different $s_N$ situation, it can be observed that an increase in the brand manufacturer's product service quality level $s_N$ reduces the AR space. This implies that if the contract manufacturer's service quality level relative to the competition is higher than that of the brand manufacturer's product, the brand manufacturer is more likely to choose the agency mode first.

It is noteworthy that the service quality level has limited impacts on the equilibrium sales mode combination of supply chain members, its impacts primarily focusing on the vicinity of the threshold $\bar{f}^A$ (on the graph in the vicinity of $f = 0.6$). In the vicinity of the threshold $\bar{f}^A$, where the brand manufacturer's profits are similar under both sales modes, the service quality level becomes the main influencing factor on the equilibrium sales mode combination. However, in regions that are not close to the threshold, the impact of the commission rate is much greater than the impact of the service quality level. When commission rate $f$ is far from the threshold $\bar{f}^A$, the commission rate becomes the primary influencing factor in the equilibrium combination, and the service quality level has no or negligible effect on the equilibrium combination, as it is outweighed by the impact of the commission rate.

Hence, it is deduced that the equilibrium sales mode combination of the brand manufacturer and the contract manufacturer is primarily contingent upon the commission rate. Specifically, when the commission rate $f$ is relatively low, the equilibrium outcome is the AA scenario, whereas a higher commission rate $f$ leads to the RR scenario as the equilibrium outcome. In this particular supply chain structure, the sales mode decisions of the supply chain members exhibit a tendency towards symmetry, and the relative competitive intensity of product service quality between both parties will affect the equilibrium sales mode combination of the brand manufacturer and the contract manufacturer near the commission rate threshold $\bar{f}^A$. More precisely, as the contract manufacturer's product service quality level demonstrates a greater relative competitive advantage, there is an increased likelihood of the equilibrium sales mode combination shifting from RR scenario to AR scenario.

## 6. Conclusions

This study, with the impact of service quality level on product demand, constructs a three-tier supply chain system consisting of a contract manufacturer, a brand manufacturer, and an e-commerce platform. The study explores the decision process of the contract manufacturer's entry into the market with its own brand and the choice of entry sales mode. The findings are as follows: (i)The contract manufacturer's decision of brand building invasion depends on

potential market demand. In addition, the potential market demand can be estimated based on the profitability of competing branded products. (ii)When the brand manufacturer adopts the reselling mode, the contract manufacturer chooses the agency mode to invade when at low commission rates and the reselling mode to invade when at high commission rates, with the commission rate threshold being constant. When the brand manufacturer adopts the agency mode, the contract manufacturer also chooses the agency mode to invade when at low commission rates and the reselling mode to invade when at high commission rates. However, the commission rate threshold is not constant and related to service quality level in this situation. (iii)When the brand manufacturer adopts the agency mode, the contract manufacturer's choice of invasion sales mode is based on the commission rate threshold, which is higher than the corresponding threshold for the contract manufacturer when the brand manufacturer adopts the reselling mode. Hence, the contract manufacturer is more likely to choose the reselling mode to invade when the brand manufacturer adopts the agency mode. (iv) When the brand manufacturer adopts the reselling mode, the service quality level has no influence on the contract manufacturer's choice of invasion sales mode. When the brand manufacturer adopts the agency mode, as the service quality level of the contract manufacturer increases, it becomes more likely to choose the reselling mode to invade for its own brand, and vice versa for the agency mode. (v) Under the condition of the contract manufacturer choosing brand building invasion, the equilibrium sales mode combination for the brand manufacturer and the contract manufacturer is as follows: at low commission rates, both parts choose the agency mode; at high commission rates, both choose the reselling mode; when the commission rate is moderate, the brand manufacturer chooses the agency mode while the contract manufacturer chooses the reselling mode.

Based on the findings of this study, the following managerial insights can be derived. First, for contract manufacturers intending to invade the online channel and establish their own brands, the key determinants of intrusion are product market demand and e- e-commerce platform commission rate. If brand manufacturers generate substantial profits from their products in the online channel, it signifies significant potential market demand, making the decision to build their own brands of contract manufacturers viable. Second, commission rates on e-commerce platforms significantly influence the sales mode decisions of both contract manufacturers and brand manufacturers, especially when commission rates are at a moderate level. In this situation, it is advisable for contract manufacturers to keep the same sales mode decisions of brand manufacturers to maximize profitability, especially that they are unsure when to alter their sales mode decisions due to uncertainty regarding the commission rate threshold. Third, contract manufacturers with high levels of service quality should consider adopting a reselling mode and entrust the sales to e-commerce platforms for their own brand products. Conversely, contract manufacturers with low service quality should opt for an agency mode and take charge of selling their products themselves. And the reason is that the product service quality level can influence the commission rate threshold (a critical parameter affecting sales mode selection).

This research could be extended in the following ways: First, this study only considers the differentiation of service quality levels and does not account for factors such as customer loyalty. Future research could explore the inclusion of customer loyalty and other relevant variables. Second, the service quality level is treated as an exogenous factor in this study. It would be beneficial to expand the analysis by treating service quality level as an endogenous variable and investigating its impact on contract manufacturers' decision-making regarding own brand intrusion and sales mode. Third, this study solely focuses on a single e-commerce platform. Future research can explore the invasion sales mode decisions of contract manufacturers in the context of multiple e-commerce platforms.

## Supporting information

**S1 Appendix. The title of supporting information files is "S1 Appendix", which contains proofs for the lemmas and propositions presented in this study, specifically, including proof materials for six lemmas and four propositions (proof of lemma 1~6, proof of proposition1~4).**
(DOCX)

## Author Contributions

**Conceptualization:** Guanxiang Zhang, Guiping Su.

**Data curation:** Xinping Liu, Guiping Su.

**Formal analysis:** Xinping Liu.

**Funding acquisition:** Guanxiang Zhang.

**Investigation:** Xinping Liu.

**Methodology:** Xinping Liu.

**Project administration:** Guanxiang Zhang.

**Resources:** Xinping Liu.

**Software:** Xinping Liu.

**Supervision:** Guanxiang Zhang, Guiping Su.

**Validation:** Xinping Liu.

**Visualization:** Xinping Liu.

**Writing – original draft:** Xinping Liu.

**Writing – review & editing:** Guanxiang Zhang, Guiping Su.

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
