## [Decision Letter · Decision Letter 0]

4 Aug 2023

PONE-D-23-21727Research on contract manufacturer invasion and sales model considering the level of service qualityPLOS ONE

Dear Dr. Su,

Thank you for submitting your manuscript to PLOS ONE. After careful consideration, we feel that it has merit but does not fully meet PLOS ONE’s publication criteria as it currently stands. Therefore, we invite you to submit a revised version of the manuscript that addresses the points raised during the review process.

We look forward to receiving your revised manuscript.

Kind regards,

Hao Guo

Academic Editor

PLOS ONE

   "This research is supported by Humanities and Social Sciences Youth Foundation,Ministry of Education of the People's Republic of China [grant number 19YJA630107],Guangdong Office of Philosophy and Social Science [grant number GD20CGL46], and 2023 Annual Project of Guangzhou Philosophy and Social Science Planning [grant number 2023GZYB20]. The funds is obtained by first author."

**Comments to the Author**

1. Is the manuscript technically sound, and do the data support the conclusions?

Reviewer #1: Yes

Reviewer #2: Partly

Reviewer #3: Partly

2. Has the statistical analysis been performed appropriately and rigorously? 

Reviewer #1: Yes

Reviewer #2: N/A

Reviewer #3: Yes

3. Have the authors made all data underlying the findings in their manuscript fully available?

Reviewer #1: Yes

Reviewer #2: Yes

Reviewer #3: Yes

4. Is the manuscript presented in an intelligible fashion and written in standard English?

Reviewer #1: Yes

Reviewer #2: No

Reviewer #3: No

5. Review Comments to the Author

Reviewer #1: This paper study the decision-making process of self-brand intrusion by CMs and the choice of sale modes. The research problem is interesting, the model is solid, and the results seems right. I appreciate the authors’ hard work. I propose the following issues, which need the authors’ concerns.

1. The Introduction is too short, such that the motivation, the scientific problem and the contribution of this paper is not clearly.

2. In Lemma 1, why the resulting $\\Pi_{C}$ and $\\Pi_{B}$ are functions of $f$? I guess this is a typo.

3. In Lemma 4, the mathematical symbols are ugly. The authors should modify them.

4. I would like to ask why the authors submit this paper to Plos One? It seems that the paper is more suitable to a business journal.

Reviewer #2: Research on contract manufacturer invasion and sales model

considering the level of service quality

This study explores the decision-making process and choice of invasion sales modes by contract manufacturers considering service quality disparities with brand manufacturers. Specifically, it constructs a three-tier supply chain system comprised of a brand manufacturer, a contract manufacturer, and an e-commerce platform. The pivotal question of whether a contract manufacturer, aspiring to build its own brand, should establish a brand and select an appropriate sales model on e-commerce platforms becomes the focus of this research. Despite the author's substantial efforts, I believe from a professional standpoint that this paper does not yet meet the publication standards of the PLOS ONE journal at this stage. Therefore, I recommend a "Major Revision" decision and provide the following specific review comments:

1. The introduction lacks an explanation of service quality and fails to articulate the paper's innovative aspects and the arrangement of its structure. The literature review section that states "this study aims to investigate the following issues" should also be included in the introduction.

2. The literature review only provides a summary of related streams, and the referenced literature is simply listed without logical connections between them. The citation format is inconsistent, and there are several inaccuracies in the page numbers of the references in the References section that need to be corrected. The review of the literature is overly broad and does not showcase the theoretical value of existing research or how this study extends and builds upon related literature, nor the theoretical contribution of this paper to the research gap.

3. The selection of parameters in the example analysis lacks necessary real-world support. The example graphs lack necessary details (such as annotation explaining thresholds), and the majority of them are analyzing the commission rate f.

4. The conclusions are overly convoluted. Among the five conclusions in this paper, only the fourth one mentions service quality. The management applications are also overly broad and do not closely relate to the previous analysis and conclusions.

Reviewer #3: The study examines the contract manufacturer's invasion against downstream market and its selling mode selection with the consideration of the differences in service quality levels. However, there are still many problems in the writing and modeling of this submission. Thus, I recommend a major revision or a resubmission.

The problems are given via an attachment word in details.

6. PLOS authors have the option to publish the peer review history of their article (what does this mean?). If published, this will include your full peer review and any attached files.

Reviewer #1: No

Reviewer #2: No

Reviewer #3: No

---

## [Author Response · Author response to Decision Letter 0]

17 Sep 2023

Dear Editor and Reviewers,

Thank you very much for giving us this opportunity to revise the manuscript.We sincerely appreciate your valuable feedback, which has been immensely helpful for our study. We have made substantial revisions and provided detailed responses to your inquiries, which can be found in the "Response to Reviewers".

With best wishes

---

## [Editor Report · Decision Letter 1]

27 Sep 2023

Manufacturer invasion and online sales mode strategy considering the level of service quality

PONE-D-23-21727R1

Dear Dr. Su,

We’re pleased to inform you that your manuscript has been judged scientifically suitable for publication and will be formally accepted for publication once it meets all outstanding technical requirements.

Kind regards,

Hao Guo

Academic Editor

PLOS ONE

---

## [Editor Report · Acceptance letter]

8 Oct 2023

PONE-D-23-21727R1 

Manufacturer invasion and online sales mode strategy considering the level of service quality 

Dear Dr. Su:

I'm pleased to inform you that your manuscript has been deemed suitable for publication in PLOS ONE. Congratulations! Your manuscript is now with our production department. 

Kind regards, 

on behalf of

Dr. Hao Guo 

Academic Editor

PLOS ONE